

# NU7441, a selective inhibitor of DNA-PKcs, alleviates intracerebral hemorrhage injury with suppression of ferroptosis in brain

Xiyu Gong[1,2], Cuiying Peng[3] and Zhou Zeng[1]

[1] Department of Neurology, The Second Xiangya Hospital, Central South University, Changsha, Hunan, China
[2] Department of Neurology, Second Hospital of Jilin University, Changchun, Jilin Province, China
[3] Department of Neurology, Hunan Provincial Rehabilitation Hospital, Hunan University of Medicine, Changsha, Hunan, China

## ABSTRACT

Neuronal apoptosis, oxidative stress, and ferroptosis play a crucial role in the progression of secondary brain injury following intracerebral hemorrhage (ICH). Although studies have highlighted the important functions of DNA-dependent protein kinase catalytic subunit (DNA-PKcs) in various experimental models, its precise role and mechanism in ICH remain unclear. In this study, we investigated the effects of DNA-PKcs on N2A cells under a hemin-induced hemorrhagic state *in vitro* and a rat model of collagenase-induced ICH *in vivo*. The results revealed a notable increase in DNA-PKcs levels during the acute phase of ICH. As anticipated, DNA-PKcs and γ-H2AX had consistent upregulations after ICH. Administration of NU7441, a selective inhibitor of DNA-PKcs, alleviated neurological impairment, histological damage, and ipsilateral brain edema *in vivo*. Mechanistically, NU7441 attenuated neuronal apoptosis both *in vivo* and *in vitro*, alleviated oxidative stress by decreasing ROS levels, and suppressed ferroptosis by enhancing GPX4 activity. These results suggest that inhibition of DNA-PKcs is a promising therapeutic target for ICH.

# INTRODUCTION

Intracerebral hemorrhage (ICH) is a life-threatening condition with high mortality and disability rates. It occurs when blood leaks into the brain tissue, forming a hematoma and compressing the surrounding tissues. Typical acute spontaneous ICH occurs in deep brain structures, including basal ganglia, thalamus, pons or deep portions of the cerebellum. While, lobar ICH presents a different clinical profile and a more severe early prognosis than subcortical ICH (*Mendiola et al., 2023*). ICH-related brain injury can be classified as primary and secondary. Primary brain injury occurs within hours of ICH and results from mechanical disruption caused by the mass effect on local brain tissue (*Lan et al., 2017*). Secondary brain injury is characterized by additional brain damage owing to

Corresponding author
Zhou Zeng,
susanzengzhou@csu.edu.cn

inflammation, disruption of the blood–brain barrier (BBB), and perihematomal edema (*Shi et al., 2020*), which are closely associated with a poor prognosis and serve as potential therapeutic targets for ICH.

Despite extensive research on primary brain injury, effective strategies aimed at controlling intensive blood pressure, maintaining hemostasis, and clearing hematomas for the treatment of ICH are lacking. Therefore, the focus of ongoing research should be shifted to elucidating the pathological mechanisms of secondary brain injury. Co-regulation factors, complement components, and cell debris from a hematoma can exert neurotoxic and immunogenic effects on the central nervous system, leading to oxidative stress; inflammatory responses; and various forms of cell death, including necrosis, apoptosis, necroptosis, pyroptosis, and ferroptosis (*Keep, Hua & Xi, 2012*; *Puy et al., 2023*; *Xi, Keep & Hoff, 2006*). In particular, hemoglobin in red blood cells generates neurotoxic reactive oxygen species (ROS) that damage cell membranes, proteins, and DNA (*Lin et al., 2023*). After DNA damage, multiple pathways are activated to initiate a complex signaling cascade known as DNA damage response (DDR), which detects DNA damage and halts cell cycle progression or promotes cell death or apoptosis in cases of severe and irreparable damage (*Ciccia & Elledge, 2010*). Because DNA is susceptible to oxidative damage, cells have evolved various DNA repair mechanisms, including base excision repair (BER), nucleotide excision repair (NER), and non-homologous end joining (NHEJ) (*Cui et al., 2000*; *Li et al., 2011*). Deficient DNA repair is closely associated with poor neurological outcomes after stroke, whereas upregulation of specific DNA repair enzymes such as APE1, OGG1, XRCC1, and Gadd45b can enhance long-term functional recovery after stroke (*Basso & Ratan, 2013*). In addition, DNA damage and repair have been shown to play a crucial role in neurogenesis, white matter repair, and neurovascular unit remodeling during functional recovery after stroke (*Blackford & Jackson, 2017*; *Wang et al., 2016*; *Zou & Elledge, 2003*).

DNA-dependent protein kinase (DNA-PK) catalytic subunit (DNA-PKcs) is a member of the phosphoinositide 3-kinase-related protein kinase (PIKK) family and a key DNA repair enzyme. It is activated mainly under stress conditions, including aging (*Park et al., 2017*), radiation (*Chen et al., 2020*), oxidative stress (*Chen et al., 2023*), asthma (*Ghonim et al., 2015*), and type 2 diabetes mellitus (*Park et al., 2017*). Under physiological conditions, DNA-PKcs primarily interacts with Ku80 (*Radhakrishnan & Lees-Miller, 2017*), which acts as a regulatory domain and participates in DNA repair by recognizing and interacting with damaged DNA fragments. Recent studies have revealed novel non-genomic functions of DNA-PKcs. For example, deficiency of DNA-PKcs has been shown to alleviate TH2-mediated airway inflammation in allergic asthma (*Mishra et al., 2015*) and DNA-PKcs induces cytochrome c release and mitochondrial apoptosis by upregulating Bax in neurodegenerative diseases (*Liu, Naegele & Lin, 2009*). However, studies investigating the role and mechanisms of DNA-PKcs in ICH are limited.

In the present study, we employed a well-established collagenase-induced rat model of intra-striatal hemorrhage to mimic typical human acute spontaneous ICH. We examined the role of DNA-PKcs in both *in vivo* and *in vitro* models of ICH. Western blotting, immunofluorescence analysis, transmission electron microscopy, and quantitative reverse

transcription polymerase chain reaction (qRT-PCR) were used to evaluate the effects of NU7441, a highly efficient and selective inhibitor of DNA-PKcs, on ICH. This study proposed a potential therapeutic strategy for ICH with inhibition of DNA-PKcs which provides novel insights into the treatment of ICH.

# MATERIALS AND METHODS

## *In vivo* experiments

### *Animals and experimental groups*

A total of 196 adult male Sprague-Dawley rats weighing 250–350 g were obtained from Hunan Slack Jingda Animal Laboratory, Changsha, China. The rats were housed in groups of three per cage in the central laboratory of Hunan Provincial People's Hospital. They were maintained on a 12-h light/12-h dark cycle at a constant temperature of 25 °C and a relative humidity level of 60%, with *ad libitum* access to water and food. Ethical approval was received from the Institutional Animal Care and Use Committee of Central South University (IACUC approval no.: 2022982). The rats were randomly divided into three experimental groups as follows: control (sham-treated, without ICH), vehicle (ICH + DMSO), and treatment (ICH + 2-mg/kg NU7441). All rats were euthanized *via* intraperitoneal injection of pentobarbital sodium.

### *Rat model of ICH*

ICH was induced through stereotactic intrastriatal injection of collagenase type IV (Sigma-Aldrich, St. Louis, MO, USA) using a previously reported method, with a few modifications (*Rehni et al., 2022*; *Zeng et al., 2017*). Briefly, SD rats were anesthetized *via* intraperitoneal administration of pentobarbital sodium (40 mg/kg). The anesthetized rats were carefully placed on a stereotactic device, and a midline incision was made to expose the bregma. A microsyringe (10 μL) was inserted into the right striatum through a burr hole at the following coordinates from the bregma: 0.5 mm anterior–posterior, 3.5 mm medial–lateral (right), and 6 mm dorsal–ventral (from pia). Collagenase IV (0.2 U) dissolved in 2-mL saline was injected gradually over 5 min. The needle was left in place for an additional 5 min and was subsequently withdrawn slowly. The body temperature of the rats was maintained at 37 ± 0.5 °C using a homeothermic blanket. Rats in the sham group underwent the same procedure; however, they received an equivalent amount of saline instead of collagenase IV.

### *Experimental design and grouping*

Part I: Temporal changes in the expression of DNA-PKcs and γ-H2AX ($n = 30$)

Western blotting (WB) was used to evaluate the protein expression of DNA-PKcs and γ-H2AX in perihematomal tissues at different time points. A total of 30 rats were randomly divided into six groups as follows: sham, ICH 6h, ICH 1d, ICH 3d, ICH 7d, ICH 14d. Three rats from each group were used for WB.

Part II: Effects of DNA-PKcs ($n = 35$)

To investigate the effects of DNA-PKcs on ICH-induced neuronal injury, rats were randomly divided into four groups as follows: sham, ICH + 0.2-U vehicle, ICH + 0.5-U vehicle, and ICH + NU7441 (2 mg/kg). Neurological scores and brain water content were

evaluated at 3 d after operation in each group. The protein expression of DNA-PKcs was analyzed *via* WB at the same time point. Histopathological changes were observed using HE staining at the same time point. Additionally, Nissl staining was used to evaluate the severity of neuronal damage at the same time point.

Part III: Mechanism of DNA-PKcs ($n$ = 48)

Rats were randomly divided into three groups as follows: sham, ICH + vehicle (0.2-U), and ICH + NU7441. WB and qRT-PCR were performed to assess the protein and mRNA expression of the ferroptosis marker GPX4, respectively. Electron microscopy was used to examine the ultrastructure of mitochondria, and IF staining was used to investigate the correlation between DNA-PKcs and GPX4. TUNEL staining was used to determine the apoptosis rate. In addition, the oxidation-sensitive fluorescent probe DCFH-DA was used to assess ROS levels in rat cells and perihematomal tissues.

### Drug treatment

NU7441 (KU-57788, Selleck, Houston, TX, USA) was initially dissolved in 4% dimethyl sulfoxide (DMSO) and subsequently mixed and clarified with polyethylene glycol 300 (PEG300), followed by the addition of Tween80. The mixture was clarified and added to ddH$_2$O. As described previously (*Zhao et al., 2006*), NU7441 was administered intraperitoneally at a dose of 2 mg/kg within 30 min of successful induction of ICH. Rats in the sham group received equal amounts of DMSO, ddH$_2$O, PEG300, and Tween80 *via* injection.

### Neurological evaluation

#### Modified neurological severity score scale

The modified neurological severity score (mNSS) scale (*Duan et al., 2022*) was used to assess behavioral deficits 72 h after the induction of ICH. Two trained researchers, who were blinded to grouping, independently calculated mNSS values. The mNSS scale includes motor, sensory, balance, and reflex tests. Neurological function was scored on a scale of 0–18 (1–6, mild impairment; 7–12, moderate impairment; 13–18, severe injury; scores of 0 and 18 represented normal performance and severe neurological deficits, respectively). With regard to the severity of neurological impairment, the inability to complete a test was scored 1, with higher scores indicating more severe neurological impairment.

### Rotarod test

The rotarod test (*Li et al., 2022*) was used to assess the motor coordination and balance of rats. Three days before modeling, for groups of SD rats for 30 min/day of training. On the third day after modeling, the rats were placed on a rod rotating at a constant speed for 5 min. The rotating speed was gradually increased after an interval of 1 h, and the time taken by the rats to fall off the rod was recorded as the test outcome.

### Corner test

The corner test (*Xu et al., 2020*) was used to evaluate the motor function and postural symmetry of rats. This test identifies and quantifies sensorimotor asymmetries, including

contralateral deficits and ipsilateral steering. For the test, two boards were placed closely together at a 30-degree angle to form a narrow corner. A rat was placed in between the boards facing the corner. When the rat reached the corner, the palps on both sides caused it to turn left or right based on the severity of neurological impairment. Rats with unilateral hemorrhagic stroke preferentially turned in the ipsilateral direction.

### Measurement of brain water content

After the rats were anesthetized, their heads were removed and brain water content was measured using the desiccation method (*Suzuki et al., 2010*). Briefly, the cerebellar tissue was removed, and the wet weights of the right and left brains were measured on an MA110 electronic analytical balance (Shanghai Second Balance Instrument Factory, Shanghai, China). The brain tissues were dried in an oven at 110 °C for 24 h, and the dry weights of the left and right brain tissues were subsequently measured. Brain water content was calculated using Elliot's formula as follows: Brain water content (%) = (wet weight – dry weight)/wet weight × 100%.

### Preparation of paraffin-embedded sections

Paraffin-embedded sections were prepared as described previously (*Yue et al., 2020*), with some modifications. After the rats were anesthetized with pentobarbital sodium, they were perfused with 250 mL of 0.01-M PBS (pH 7.4) *via* the aorta, followed by the addition of 500 mL of 4% paraformaldehyde. The brain tissue was removed and fixed in 4% paraformaldehyde at 4 °C for 24–48 h. After dehydration and vitrification, the tissue specimens were embedded in paraffin and cut into 4-μm-thick sections. The sections were dewaxed in xylene and rehydrated in a graded series of ethanol solutions and deionized water. These paraffin-embedded tissue sections were used for subsequent IF analysis, HE staining, TEM, Nissl staining, and TUNEL immunostaining.

### Hematoxylin and eosin staining

The rats were perfused with 4% paraformaldehyde and decapitated. The brains were removed, fixed in 4% paraformaldehyde for 1 day, embedded in paraffin, and cut into 4-μm-thick sections. The tissue sections were flushed with tap water and stained with hematoxylin for 5 min. The sections were differentiated using 0.1% hydrochloric acid and ethanol solution for 25 s, immersed in phosphate-buffered saline for 45 min, and counterstained with eosin for 5 min. The sections were subsequently dehydrated in 95% ethanol and mounted with neutral balsam. Hematomas were identified by the presence of intact or lysed red blood cells. Perihematomal images were captured using an optical microscope (Nikon Eclipse E100).

### IF staining

IF staining was performed as described previously (*Alikarami et al., 2017*), with some modifications. Briefly, 4-μm-thick paraffin-embedded sections were subjected to antigen retrieval and blocked with 5% BSA for 40 min. The sections were incubated with anti-DNA-PKcs antibody (ab32566; Abcam, Cambridge, UK) overnight at 4 °C. The following day, the sections were washed with PBS and incubated with an anti-GPX4 antibody

(67763-1-Ig; Proteintech, Rosemont, IL, USA) for 1 h at room temperature. After three PBS washes, the sections were stained with 4′6-diamidino-2-phenylindole (DAPI) for 10 min, and images were captured under a fluorescence microscope (ZEISS-AXIO Scope Al, Oberkochen, Germany).

### qRT-PCR

qRT-PCR was used to verify the expression of DNA-PKcs and GPX4 genes in three groups of N2A cells. The total RNA of N2A cells was extracted by Trizol (MRC TR118-500) method according to the instructions, and the concentration of each RNA sample was detected. High-quality RNA typically exhibits an A260/A280 ratio of 1.8–2.0. Approximately 2 μL of each RNA sample at 40 μg/mL was reverse transcribed into cDNA using M-MLV Reverse Transcriptase (Promega M1705; Promega, Madison, WI, USA) for mRNA qRT-PCR. Relative expression was quantified using the $2^{-\Delta\Delta CT}$ method and normalized to the control group. The detection protocol included a denaturation temperature of 95 °C, an annealing temperature of 60 °C, and an extension temperature of 72 °C.

The primer sequences for qRT-PCR are listed in Tables 1–3.

### TUNEL immunostaining

Terminal deoxynucleotidyl transferase-mediated dUTP nick-end labeling (TUNEL) detected apoptosis in the ipsilateral cortical neurons of SD rats. Brain tissues from each group were incubated with TUNEL reaction mixture (Beyotime, Shanghai, China) for 1 h at room temperature and subsequently stained with DAPI (Wellbio, Qingdao, China). The tissues were then examined using a fluorescence microscope (Motic, Nanjing, China).

### Electron microscopy

Specimens for electron microscopy were prepared as described previously (*Fan et al., 2016*; *Rahal et al., 2016*) and observed using a Hitachi HT7700 transmission electron microscope (Tokyo, Japan). All tissue sections were analyzed at the same intensity and magnification.

### *In vitro* experiments

### Cell culture and processing

The mouse adult neuroblastoma cell line N2A (CCL-131, ATCC; RRID: CVCL_0470) was cultured in DMEM (SH30022.01; Hyclone, Logan, UT, USA) supplemented with 10% fetal bovine serum (10099141; Gibco, Grand Island, NY, USA) and 1% penicillin–streptomycin (AWH0529a; Abiowell, Boulder, CO, USA) and maintained at 37 °C in a humidified environment containing 5% $CO_2$. The culture medium was changed every 2–3 days. After the cells reached approximately 80% confluence, they were digested with trypsin (C0201; Beyotime, Seoul, Korea) and divided into two groups for passaging.

### N2A cell model of ICH

Cells in the logarithmic growth phase were seeded in a 6-well plate and divided into the following three groups: sham, hemin, and hemin + NU7441. The three groups of cells were spiked at different time points so that the three groups of cells were treated on the same day. Cells in the sham group were cultured normally, those in the hemin group were

**Table 1 Reaction system.** For organization total mRNA templates, reverse transcription cDNA, reaction system as follows.

| Reagent | 20 ul reaction system |
| --- | --- |
| dNTP mix, 2.5 mM each | 4 uL |
| Primer mix | 2 uL |
| RNA template | 7 uL |
| 5×RT buffer | 4 uL |
| DTT, 0.1 M | 2 uL |
| HiFiScript, 200 U/uL | 1 uL |

**Table 2 qPCR composition system.** Real time quantitative PCR reagents and volume.

| Reagent | Volume |
| --- | --- |
| Template | 2 uL |
| Primer R (10 uM) | 1 uL |
| Primer F (10 uM) | 1 uL |
| ddH$_2$O | 11 uL |
| 2X SYBGREEN PCR master mix | 15 uL |

**Table 3 qPCR primer design.** Primer gene sequence and length.

| | Forward primer | Reverse primer | Product length |
| --- | --- | --- | --- |
| DNAPKcs | ACCAGAGAGCATTCCATCACC | GAGTTGTTGGTCACAGAAGCC | 140 bp |
| GPX4 | CATCGACGGGCACATGGTCT | CCACACTCAGCATATCGGGCAT | 138 bp |
| Actin | ACATCCGTAAAGACCTCTATGCC | TACTCCTGCTTGCTGATCCAC | 223 bp |

treated with 10-uM hemin for 48 h, and those in the hemin + NU7441 group were treated initially with 1-uM NU7441 for 24 h and subsequently with 10-uM hemin for 48 h.

### Drug administration

N2A cells were pre-treated with NU7441 dissolved in DMSO and subsequently incubated with 10-mM hemin for 24 h. Cells in the vehicle group were treated with an equal volume of PBS.

### CCK-8 assay

Cell viability was assessed using Cell Counting Kit-8 (CCK-8; Dojindo Molecular Technologies, Dojindo, Japan) according to the manufacturer's protocol. After 1 h of incubation with CCK-8 reagent at 37 °C, the optical density (OD) of cells in each well was measured at 450 nm using an enzyme-linked immunosorbent device.

### Assessment of apoptosis via flow cytometry

After the culture medium was discarded, N2A cells were harvested using EDTA-free trypsin and washed twice with PBS. The cell suspension was centrifuged to obtain approximately $3.2 \times 10^5$ cells. These cells were treated with a binding buffer (500 mL) and incubated with annexin V-APC (5 mL) for 10 min. Finally, the cells were examined using an FACS C6 flow cytometer (FCM; FACSCanto II; BD Biosciences, Franklin Lakes, NJ, USA), and the results were analyzed using the FlowJo (v10.7.1) software (Tree Star, Ashland, OR, USA).

### Western blotting

Rat perihematoma brain tissues and N2A cells were processed for WB as described previously (*Wu et al., 2022*). Anti-γ-H2AX (ab126171; Abcam, Cambridge, UK), anti-DNA-PKcs (ab32566; Abcam, Cambridge, UK), anti-GPX4 (67763-1-Ig; Proteintech, Rosemont, IL, USA), and anti-β-actin (60008-1-Ig; Proteintech, Rosemont, IL, USA) were used as primary antibodies, whereas anti-rabbit IgG and anti-mouse IgG (Proteintech, Rosemont, IL, USA) were used as secondary antibodies. Protein bands were visualized using an enhanced chemiluminescence (ECL) detection kit (Advansta Inc., San Jose, CA, USA).

## Statistical analysis

All data were expressed as the mean ± standard error of the mean (SEM). The SPSS Statistics (version 20.0; Armonk, NY, USA) software was used for statistical analysis. Student's t-test was used to compare data between two groups, whereas one-way or two-way analysis of variance (ANOVA) followed by Bonferroni *post hoc* test was used to compare data among multiple groups. A *p*-value of <0.05 was considered statistically significant.

## RESULTS

### Expression of DNA-PKcs and γ-H2AX after the induction of ICH *in vivo* and *in vitro*

To determine the role of DNA damage in SBI after ICH, WB was used to evaluate the protein expression of DNA-PKcs and γ-H2AX in both *in vitro* and *in vivo* models of ICH. The results showed that the expression of both proteins in ipsilateral brain homogenates of rats with collagenase-induced ICH increased after 1 day of modeling and peaked on day 3. Thereafter, the expression of both proteins showed a gradual and persistent decrease, eventually returning to normal on day 7. On day 14, no difference was observed in the expression of both proteins between the ICH and sham groups (Figs. 1B–1D). Furthermore, WB and qRT-PCR showed that the protein and mRNA expression of DNA-PKcs and γ-H2AX were significantly increased in N2A cells treated with hemin for 48 h (Fig. 2). The expression of DNA-PKcs increases with the aggravation of brain injury (Figs. 1E–1G). These results suggest that DNA-PKcs play an important role in SBI caused by acute ICH.

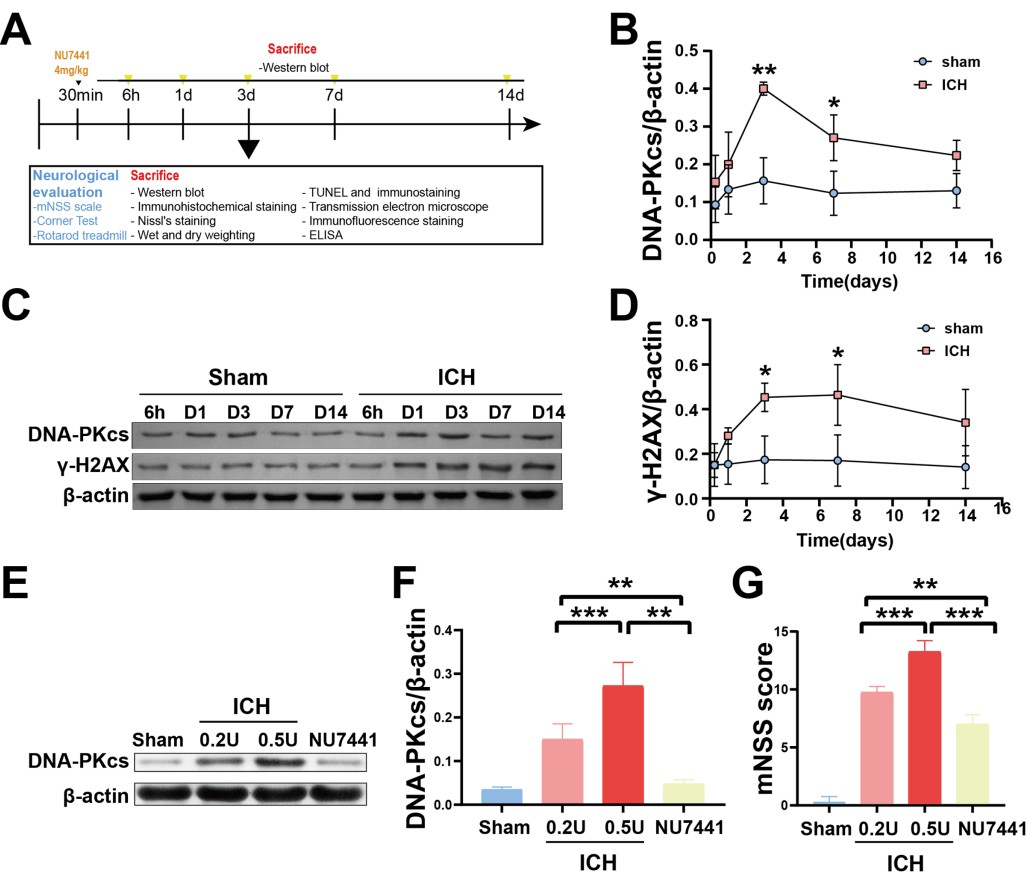

**Figure 1** **Temporal changes in the protein expression of DNA-PKcs and γ-H2AX in rats during the acute phase of intracerebral hemorrhage.** A simplified depiction of the experimental design is shown in (A). Western blotting (C) and quantification of DNA-PKcs (B) and γ-H2AX (D) in a rat model of ICH. Western blotting (E) and quantification (F) of DNA-PKcs in rats with 0.2-U or 0.5-U collagenase-induced ICH. mNSS score (G). All data are expressed as the mean ± standard deviation ($*p < 0.05$; $**p < 0.01$; $***p < 0.001$).

## Neuroprotective effects of DNA-PKcs inhibition *in vivo* and *in vitro*

The results of mNSS scale, corner test, and rotarod test showed that rats had marked neurological dysfunction on day 3 after the induction of ICH (Figs. 2G, 3A, 3B). Administration of NU7441 significantly alleviated behavioral deficits on day 3 after ICH. Consistent with the results of the behavioral tests, H&E and Nissl staining showed that NU7441 alleviated tissue damage and reduced neuronal apoptosis (Figs. 3C–3E). Compared with the NU7441 group, the ICH group showed more severe tissue edema, with a larger number of highly pigmented cells with nuclear atrophy and a smaller number of Nissl bodies. Furthermore, CCK-8 assay was used to examine the effects of NU7441 on the viability of N2A cells *in vitro*. The selective inhibition of DNA-PKcs by NU7441 significantly increased the survival rate and density of hemin-treated N2A cells (Figs. 3F and 3G).

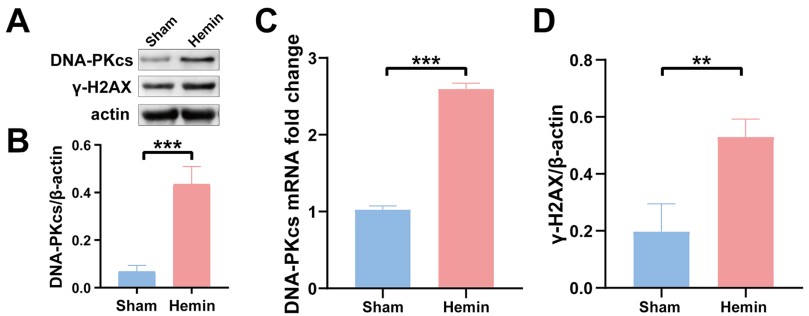

**Figure 2 Protein and mRNA expression of DNA-PKcs and γ-H2AX in N2A cells treated with hemin for 48 h.** Western blotting (A) and qRT-PCR (C) were performed to evaluate the protein and mRNA expression of DNA-PKcs (B) and γ-H2AX (D) in N2A cell model of ICH, respectively. All data are expressed as the mean ± standard deviation (**$p < 0.01$; ***$p < 0.001$).

## Inhibition of DNA-PKcs alleviated ICH-induced brain edema

Compared with the sham group, the ICH group had significantly higher BWC in the ipsilateral basal ganglia on day 3 after the induction of ICH ($p < 0.001$, Fig. 4). On the contrary, NU7441 significantly decreased BWC, suggesting that inhibition of DNA-PKcs played an important role in reducing brain edema (vector *vs.* NU7441; $p < 0.001$).

## Inhibition of DNA-PKcs reduced ICH-induced neuronal apoptosis *in vivo* and *in vitro*

TUNEL staining was performed to determine the number of apoptotic neurons in the perihematomal tissues of rats with ICH (Figs. 5A and 5B). Only a few apoptotic neurons were observed in the sham group. As shown in Fig. 6B, the number of apoptotic neurons was substantially increased in the ICH group but significantly decreased in the NU7441 group ($p < 0.001$). To validate the inhibitory effects of NU7441 on apoptosis *in vitro*, flow cytometry was used to quantify apoptotic cells. Treatment with hemin promoted the apoptosis of N2A cells; however, treatment with NU7441 suppressed the hemin-induced apoptosis ($p < 0.001$, Figs. 5C and 5D).

## Inhibition of DNA-PKcs alleviated ICH-induced oxidative stress *in vivo* and *in vitro*

To evaluate the effects of DNA-PKcs on cellular oxidative stress after ICH, ROS levels in rat brain tissues were measured using the DCFH-DA probe on day 3 after the induction of ICH (Figs. 6A and 6C). ROS levels in perihematomal tissues were significantly higher in the ICH group than in the sham group ($p < 0.01$) but significantly lower in the NU7441 group than in the ICH group ($p < 0.05$). Similar results were obtained *in vitro* (Figs. 6B and 6D).

## Inhibition of DNA-PKcs suppressed ICH-induced ferroptosis *in vivo* and *in vitro*

TEM was used to assess ferroptosis after the induction of ICH. As shown in Figs. 7C–8E, both rat brain tissues and N2A cells exhibited mitochondrial shrinkage, an increased

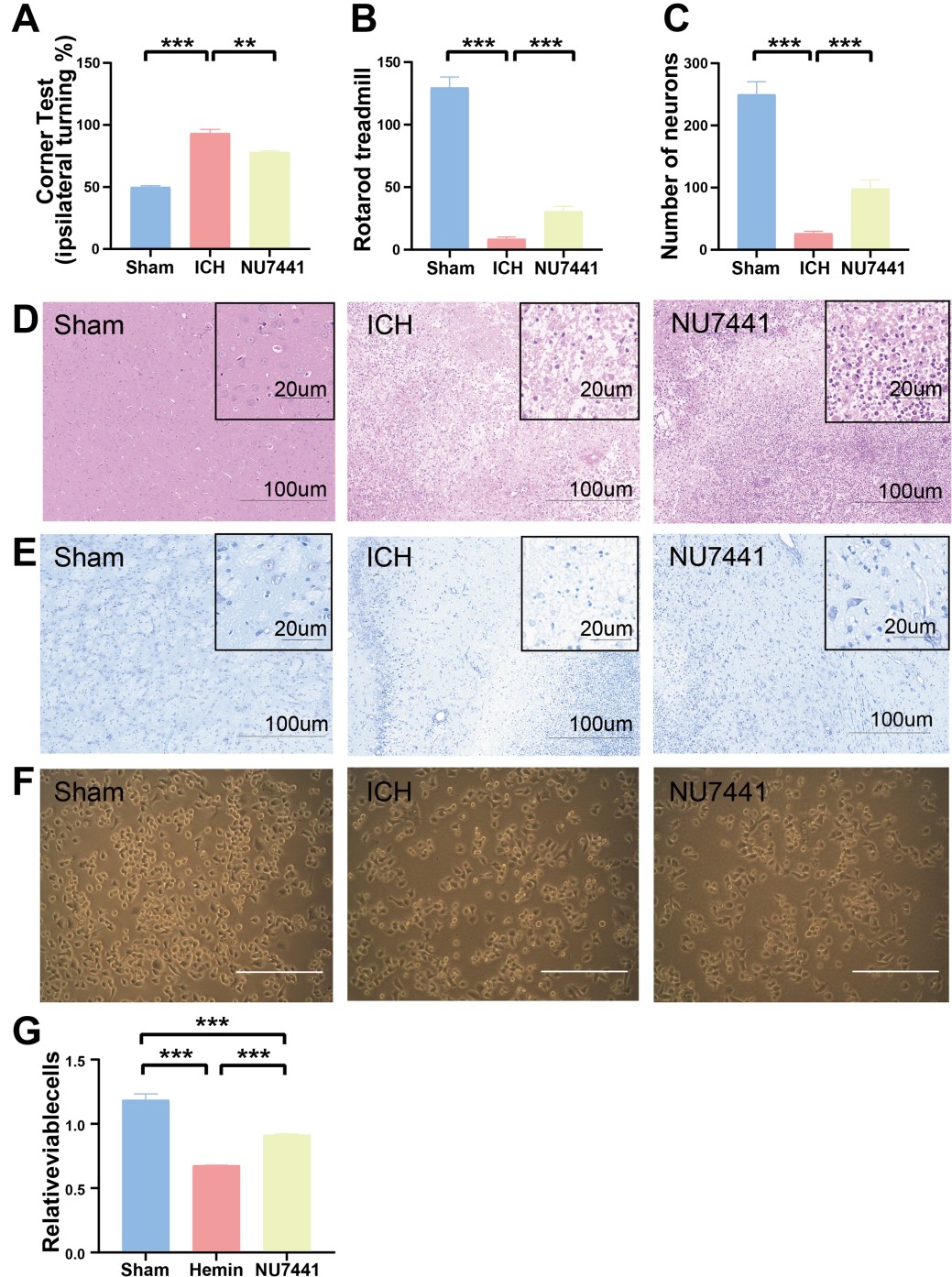

**Figure 3** **Effects of DNA-PKcs inhibition on brain injury caused by ICH.** (A and B) The neurological function of rats was assessed using the rotarod test, and corner test 3 days after the induction of ICH. (C–E) Histological changes were evaluated using H&E and Nissl staining on day 3 after the induction of ICH in rats (scale bar = 100 and 20 μm). (F and G) N2A cells were treated with hemin (10 mM) for 48 h, and their viability was assessed using light microscopy and CCK8 assay. All data are expressed as the mean ± standard deviation (**$p < 0.01$; ***$p < 0.001$).

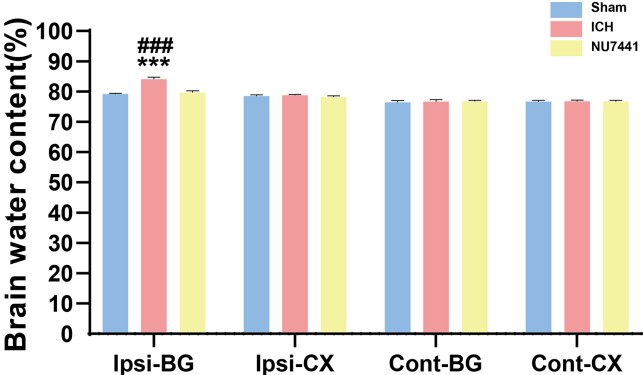

**Figure 4 Brain water content was evaluated based on the dry-to-wet weight ratio on day 3 after the induction of ICH.** Cont, contralateral; Ip, ipsilateral; BG, basal ganglia; CX, cortex; cerebel, cerebellum. (***$p < 0.001$ *vs.* the sham group) (###$p < 0.001$ *vs.* the NU7441 group).

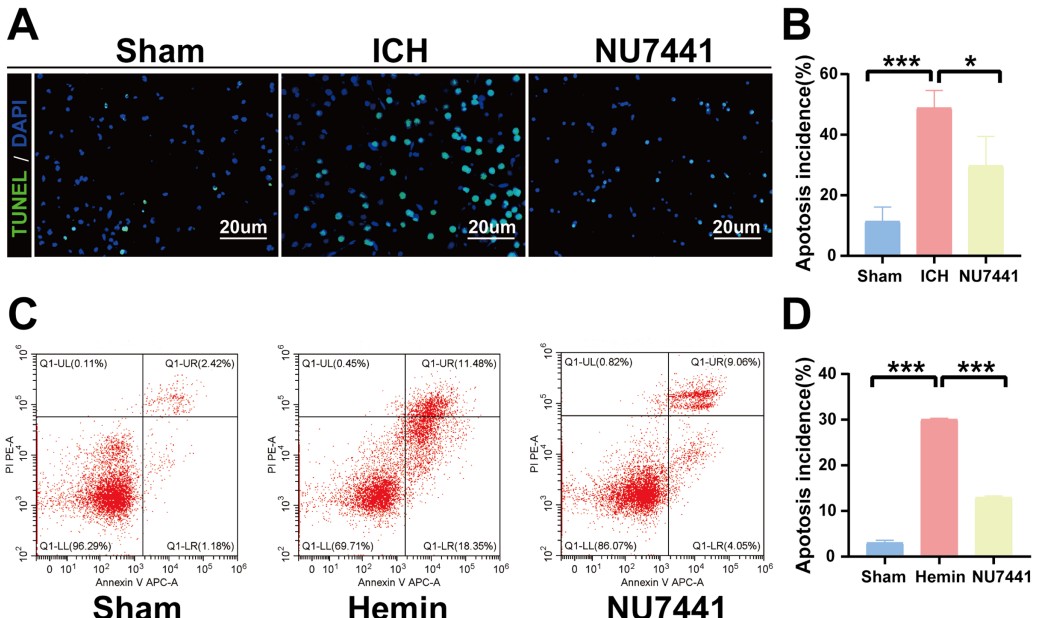

**Figure 5 Inhibition of DNA-PKcs attenuated apoptosis after ICH.** (A and B) The apoptosis rate was assessed through TUNEL staining on day 3 after the induction of ICH in rats (scale bar = 20 μm). (C and D) N2A cells were treated with hemin for 48 h, and cell apoptosis was detected *via* flow cytometry. All data are expressed as the mean ± standard deviation (*$p < 0.05$; ***$p < 0.001$).

membrane density, and outer membrane rupture, indicating typical ferroptosis. However, treatment with NU7441 decreased the intracellular mitochondrial membrane density and the bilayer membrane density. Furthermore, the protein expression of the ferroptosis marker GPX4 was evaluated *via* WB. As shown in Figs. 7A and 7B, the protein expression of GPX4 in rat perihematomal brain tissues was lower in the ICH group than in the sham group ($p < 0.001$) but significantly higher in the NU7441 group than in the ICH group ($p < 0.001$). The results of WB and qRT-PCR *in vitro* are shown in Figs. 8A–8C. In

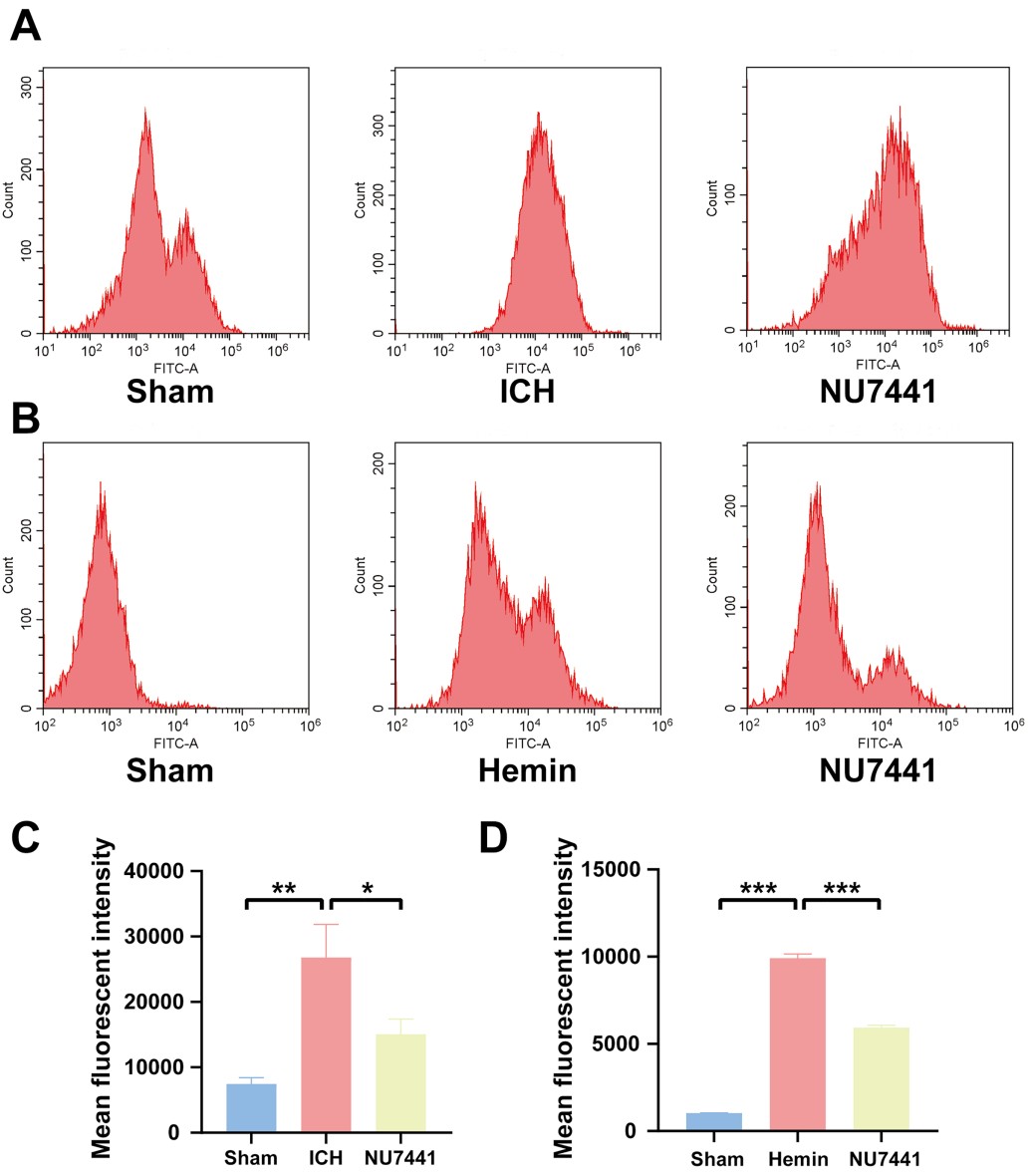

**Figure 6** **Inhibition of DNA-PKcs suppressed oxidative stress after ICH.** (A and C) ROS levels in perihematomal brain tissues were measured *via* flow cytometry on day 3 after the induction of ICH in rats. (B and D) After 48 h of hemin treatment, flow cytometry was used to measure ROS levels in N2A cells. All data are expressed as the mean ± standard deviation (*$p < 0.05$; **$p < 0.01$; ***$p < 0.001$).

particular, the protein and mRNA expression of GPX4 were lower in the hemin group than in the sham group ($p < 0.001$) but higher in the hemin + NU7441 group than in the hemin group ($p < 0.01$). Subsequently, immunofluorescence co-localization was performed to verify the interaction between DNA-PKcs and GPX4, and the results are shown in Fig. 7. In particular, DNA-PKcs and GPX4 were found to be co-localized in the hemin group (Fig. 8D).

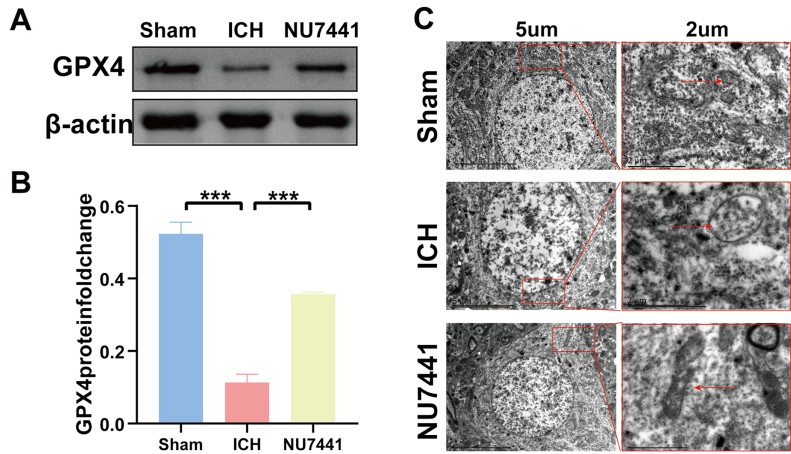

**Figure 7 Inhibition of DNA-PKcs suppressed ferroptosis after ICH *in vivo*.** (A and B) Western blotting and quantification analysis of GPX4 on day 3 after the induction of ICH in rats. (C) Transmission electron microscopy was used to examine mitochondrial changes in rat brain tissues after 3 days of ICH induction (scale bar = 2 and 5 μm). All data are expressed as the mean ± standard deviation (***$p < 0.001$).

## DISCUSSION

In this study, we investigated the role of DNA-PKcs in brain damage following ICH and elucidated the underlying pathological mechanism. The key findings of this study are as follows: (1) DNA-PKcs was upregulated in rat models of collagenase-induced ICH; specifically, the expression of DNA-PKcs began to increase gradually from 6 h after ICH, peaked on day 3 after ICH, continued to increase until day 7 after ICH, and returned to normal by day 14 after ICH. Notably, the expression pattern of the DSB marker γ-H2AX was similar to that of DNA-PKcs. In addition, the expression of DNA-PKcs was correlated with the severity of brain injury. Consistently, *in vitro* experiments showed that treatment with hemin significantly increased the expression of DNA-PKcs in N2A cells after 48 h (2) Inhibition of DNA-PKcs using NU7441 alleviated ICH-induced neurological deficits, histological damage, and brain edema in rats *in vivo* and increased the viability of N2A cells treated with hemin for 48 h. (3) Both animal and cell experiments validated that inhibition of DNA-PKcs exerted neuroprotective effects by reducing peripheral neuronal apoptosis, decreasing ROS levels, and preventing ferroptosis. Altogether, these results suggest that targeting DNA-PKcs is a promising therapeutic strategy for ICH.

DNA-PKcs is involved in various pathways, including stress response regulation, cellular apoptosis, telomere homeostasis, and specific gene transcription. For example, ROS-induced DNA damage can activate DNA-PKcs and its downstream signaling (*Li et al., 2014*). Studies have shown that DNA-PKcs is associated with the protein complex involved in base excision repair, which is crucial for repairing oxidative base damage (*Parlanti et al., 2007*). DNA damage includes base damage, formation of pyrimidine dimers, and single- or double-strand breaks. The DNA-PK complex primarily participates in the repair of double-strand breaks (DSBs). DSBs pose a serious threat to genome stability and cell fate and may lead to cell stasis, impairment of cellular function, and apoptosis, thereby triggering major diseases (*Li et al., 2014*; *Parlanti et al., 2007*). After

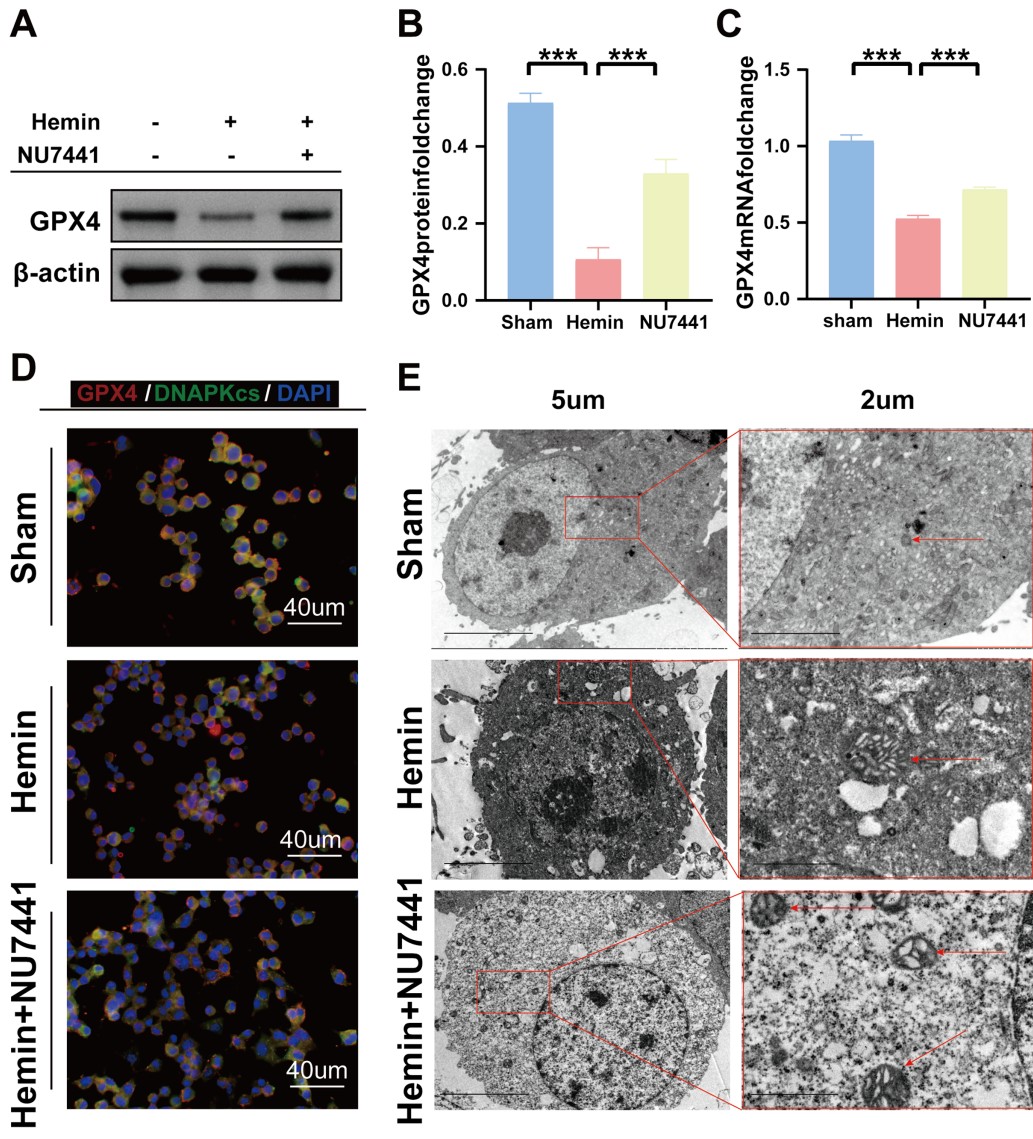

**Figure 8 Inhibition of DNA-PKcs suppressed ferroptosis after ICH *in vitro*.** (A–C) Western blotting and qRT-PCR were used to detect the protein and mRNA expression of GPX4 in N2A cells treated with hemin for 48 h, respectively. (D) Immunofluorescence analysis was performed to assess the co-localization of GPX4 and DNA-PKcs. (E) Transmission electron microscopy was used to examine the morphological characteristics of mitochondria. All data are expressed as the mean ± standard deviation (***$p < 0.001$).

DSBs occur, DNA-PKcs, the core enzyme of the DNA-PK complex, is recruited to the site of damage for activation. Activated DNA-PKcs undergoes mutations at multiple sites and phosphorylates target proteins, participating in a series of DNA damage response and repair processes and consequently determining cell fate. Love et al. showed that the expression of DNA-PKcs in neurons surrounding the infarct site increased during the acute phase of human ischemic stroke but rapidly decreased after reperfusion on day 2. In the late stage of cerebral infarction, weak expression of DNA-PKcs was observed in only neovascularized tissues and some mononuclear inflammatory cells (*Wang et al., 2018a*).

However, surviving neurons at the infarct site did not exhibit immunoreactivity to DNA-PKcs. In the first part of this study, we investigated the expression of DNA-PKcs and the DSB marker γ-H2AX in rat models of collagenase-induced ICH and N2A cell models of hemin-induced ICH through WB and qRT-PCR. The results revealed an increase in the expression of DNA-PKcs and γ-H2AX during the acute phase of ICH. Additionally, the expression of DNA-PKcs was positively correlated with the severity of brain injury. Although DNA-PKcs was found to be upregulated after ICH, its significance in the study remains unclear. Therefore, in the second part of this study, we examined the neuroprotective effects of DNA-PKcs inhibition on rat and N2A cell models of ICH. The results showed that the DNA-PKcs inhibitor NU7441 alleviated ICH-induced neurological deficits and ipsilateral brain edema in the acute phase. H&E and Nissl staining validated that inhibition of DNA-PKcs alleviated histopathological damage and reduced neuronal death after ICH. Consistently, *in vitro* experiments showed that inhibition of DNA-PKcs enhanced the survival of hemin-treated N2A cells. To the best of our knowledge, this study is the first to demonstrate the neuroprotective role of DNA-PKcs inhibition in cell and animal models of ICH. However, the underlying mechanisms warrant further investigation.

Apoptosis is the main mechanism underlying early tissue damage in the perihematomal region after ICH. Therefore, suppressing apoptosis can alleviate brain tissue damage and play a neuroprotective role in ICH (*Hosseinzadeh et al., 2016*; *Zhen et al., 2016*). Several studies have demonstrated the role of DNA-PKcs in apoptosis; however, the effects of DNA-PKcs on apoptosis vary across models. In a mouse model of sepsis-induced multi-organ failure, knockout of DNA-PKcs has been shown to reduce LPS-induced mitochondrial oxidative stress and apoptosis, consequently improving liver function (*Zou et al., 2022*). In addition, *Liu, Naegele & Lin (2009)* showed that DNA-PKcs in red algae glycine Bax induction of apoptosis signaling pathways play a necessary role. Activation of DNA-PKcs through simulated microgravity is associated with the apoptosis of human promyelocytic leukemia cells, and the underlying mechanism involves ROS overproduction and Bax activation (*Singh, Rajput & Singh, 2021*). However, a study showed that co-treatment with NU7441 and doxorubicin promoted the apoptosis of B-cell precursor acute lymphoblastic leukemia cells, leading to a decrease in DNA damage repair measured *via* γ-H2AX focus formation, which resulted in the loss of AIM2 inflammasomes in melanoma lesions and limited the activation of DNA-PKcs. These findings indicated that it suppressed Akt activation and promoted cell apoptosis (*Wilson et al., 2015*). In addition, the number of TUNEL-positive cells in hypoxic rat pulmonary artery smooth muscle cells (PASMCs) was higher in the NU7026 (a DNA-PKcs inhibitor) group than in the control group. Another study showed that NF-κB, a classical activator of cellular responses to inflammation and DNA damage, alleviated genotoxic stress and suppressed apoptosis in DNA-PKcs-deficient cells (*Medunjanin et al., 2020*). Similarly, a study on mouse models of ovalbumin-induced asthma demonstrated that the DNA-PKcs inhibitor NU7441 attenuated the inflammatory response in lung airway cells, highlighting the crucial role of DNA-PKcs in regulating asthma-induced inflammation (*Wang et al., 2018b*). In this study, treatment with NU7441 decreased the number of TUNEL-positive

cells in the perihematomal brain tissues of rats with ICH and reduced the apoptosis rate of hemin-treated N2A cells. These findings suggest that inhibiting DNA-PKcs can prevent neuronal apoptosis after ICH.

Oxidative stress is caused by ROS accumulation after ICH and plays an important role in inflammatory responses, apoptosis, autophagy, and secondary brain injury caused by BBB disruption (*Cadet et al., 2012*). After ICH, various cellular components promote ROS production, which causes damage to proteins, lipids, and nucleic acids and induces inflammation, autophagy, apoptosis, and BBB disruption (*Yao, Bai & Wang, 2021*). Therefore, intracellular ROS levels can directly reflect the severity of oxidative stress. Previous studies have suggested that DNA-PKcs directly interacts with XRCC1 to alleviate oxidative DNA damage *via* the BER pathway (*Lévy et al., 2006*). In this study, inhibition of DNA-PKcs decreased ROS levels in both *in vitro* and *in vivo* models of ICH, suggesting that inhibiting DNA-PKcs can alleviate cellular oxidative stress after ICH.

Ferroptosis is a non-apoptotic form of cell death (*Cui et al., 2021*) characterized by iron-dependent accumulation of toxic lipid ROS, which can cause irreparable lipid damage and increase membrane permeability, eventually resulting in membrane destruction and non-specific membrane perforation (*Imai et al., 2017*). Ferroptosis can be induced by inhibiting the cystine/glutamate antiporter system or decreasing the activity of glutathione peroxidase 4 (GPX4) (*Friedmann Angeli et al., 2014*). Morphologically, ferroptosis is associated with the shrinkage of mitochondria and enlargement of cristae (*Dixon et al., 2012*; *Gatti et al., 2020*). Hemolysis in cerebral hematoma produces iron ions, which can form highly toxic hydroxyl radicals that attack DNA, protein, and lipid membranes; impair cell functions, and lead to neuronal apoptosis (*Li et al., 2017*). Recent studies have shown that pterostilbene, a natural antioxidant, can suppress ferroptosis and alleviate damage in high-dose-fructose–stimulated glomerular podocytes by downregulating SSBP1 and inhibiting the DNA-PK/p53 pathway (*Wu et al., 2022*). In this study, inhibition of DNA-PKcs increased the expression of GPX4 protein in both *in vivo* and *in vitro* models of ICH. In addition, TEM showed that inhibition of DNA-PKcs reduced mitochondrial shrinkage and increased the mitochondrial membrane density. These results suggest that inhibition of DNA-PKcs plays a positive role in attenuating ferroptosis after ICH.

## Limitations of this study

DNA-PKcs is crucial for brain development and plays a significant role in preventing endogenous DNA damage. This study investigated the potential role of NU7441, a highly selective DNA-PKcs inhibitor, in mitigating secondary brain injury following ICH. Our findings indicate a dual role of DNA-PKcs in brain tissues. Selective inhibition of DNA-PKcs by NU7441 demonstrated neuroprotective effects by suppressing neuronal apoptosis and ferroptosis, as well as alleviating oxidative stress in both *in vivo* and *in vitro* models of ICH. However, the regulatory pathway of DNA-PKcs underlying ferroptosis has not yet examined in depth, as specific mechanisms were not included due to the lack of relevant inhibitors. In the future, we plan to establish a stable cell line with DNA-PKcs knockdown, and conduct transcriptome analysis and biological experiments to further elucidate its regulatory pathway. Especially, DNA-PKcs related molecular mechanisms, by

which neuronal organelles, such as Golgi apparatus and mitochondria, alter their morphology and functions to induce ferroptosis after ICH. Acute spontaneous ICH in deep brain structures, primarily attributed to small vessel diseases such as hypertensive arteriopathy and cerebral amyloid angiopathy, accounts for 85% of all ICH cases. In contrast, hematological disorders (acquired, iatrogenic, or congenital), vascular malformations, neoplasms, infections, drug abuse, and other risk factors are also recognized as important causes of lobar ICH, often requiring different therapeutic strategies, with varying risks of recurrence and outcomes (*Arboix & Besses, 1997*; *Gatti et al., 2020*; *Kaiser et al., 2019*; *Planton et al., 2020*). In this study, we mainly identified the effect and mechanism of DNA-PKcs as a therapeutic target in the treatment of intra-striatal hemorrhage. Future research will further explore the therapeutic effects of DNA-PKcs in lobar ICH arising from diverse etiologies.

## CONCLUSIONS

This study demonstrates that DNA-PKcs plays a crucial role in the pathological progression of brain injury after ICH. Inhibition of DNA-PKcs can effectively suppress apoptosis and ferroptosis and alleviate oxidative stress, thereby preventing the development of SBI after ICH. These findings strongly suggest that targeted inhibition of DNA-PKcs is a promising therapeutic strategy for ICH.

### Funding

This study was financially supported by the Natural Science Foundation of Hunan Province (Grant no.: 2023JJ40810). The funders had no role in study design, data collection and analysis, decision to publish, or preparation of the manuscript.

### Grant Disclosures

The following grant information was disclosed by the authors:
Natural Science Foundation of Hunan Province: 2023JJ40810.

### Competing Interests

The authors declare that they have no competing interests.

### Author Contributions

- Xiyu Gong conceived and designed the experiments, performed the experiments, analyzed the data, prepared figures and/or tables, authored or reviewed drafts of the article, and approved the final draft.
- Cuiying Peng performed the experiments, analyzed the data, prepared figures and/or tables, and approved the final draft.
- Zhou Zeng analyzed the data, authored or reviewed drafts of the article, and approved the final draft.

## Animal Ethics

The following information was supplied relating to ethical approvals (*i.e.*, approving body and any reference numbers):

All experimental procedures were approved by the Institutional Animal Care and Use Committee of Central South University (IACUC approval no.: 2022982).

## Data Availability

The raw measurements are available in the Supplemental Files.

## Supplemental Information

Supplemental information for this article can be found online at http://dx.doi.org/10.7717/peerj.18489#supplemental-information.

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
