# Peer review of "NU7441, a selective inhibitor of DNA-PKcs, alleviates intracerebral hemorrhage injury with suppression of ferroptosis in brain"

_PeerJ, doi:10.7717/peerj.18489_

## Round 0.1 · original submission · Major Revisions

Following evaluation of your manuscript, the reviewers have highlighted several aspects for improvement, on both mechanistic grounds and in the interpretation and wider validity of the work. We look forward to your detailed response to reviewers.

·

Basic reporting

The authors present the results of an experimental study aimed to investigate the effects of DNA-PKcs in N2A cells in a hemin-induced hemorrhagic state in vitro and in a rat model of collagenase-induced intracerebral hemorrhage (ICH) in vivo. The authors observed a marked increase in DNA-PKcs levels during the acute phase of ICH. Administration of NU7441, a selective inhibitor of DNA-PKcs, alleviated neurological impairment, histological damage, and ipsilateral brain edema in vivo. Mechanistically, NU7441 attenuated neuronal apoptosis both in vivo and in vitro, alleviated oxidative stress by decreasing ROS levels, and suppressed ferroptosis by enhancing GPX4 activity. The study is potentially interesting, but can improved if the following considerations are addressed:

Experimental design

-Original primary research is within aims and scope of the journal.

-Methods are described with sufficient detail

Validity of the findings

-The authors should mention in the Introduction that acute spontaneous lobar cerebral hemorrhages in humans present a different clinical profile and a more severe early prognosis than deep subcortical intracerebral hemorrhages (Biomedicines 2023 Jan 16;11(1):223) and should add that non-hypertensive mechanisms of intracerebral hemorrhages predominate in the lobar location. Did the authors take this into account in their study protocol?

-It would be interesting to include in the Discussion a comment on the fact that in humans acute intracerebral hemorrhage can be the presenting manifestation of a hematological disorder. This is a noteworthy aspect that should be highlighted. Emphasis should also be placed on the need to distinguish hematologic disorders from other hemorrhagic stroke etiologies that have a different treatment approach, risk of recurrence and outcome (Eur Neurol (1997) 37 (4): 207–211). Add and comment on the reference.

Additional comments

-It would be convenient to add a comment on the limitations of this study

-A brief concluding comment on other possible lines of future research on the presented topic would be appreciated

Reviewer 2 ·

Basic reporting

1. Title "Role of DNA-PKcs in secondary brain injury following intracerebral hemorrhage" is not optimal because it does not reflect effects in detail.
2. the indicators do not contain detailed mechanisms because the inhibitor of pathway is not in use.
3. Administration of NU7441, a selective inhibitor of DNA-PKcs, alleviated neurological impairment, histological damage, and ipsilateral brain edema in vivo. Mechanistically, NU7441 attenuated neuronal apoptosis both in vivo and in vitro, alleviated oxidative stress by decreasing ROS levels, and suppressed ferroptosis by enhancing GPX4 activity. These contents only indicate that it is detrimental in acute brain injury.
Thus, the study is devoid of enough depth.

Experimental design

1. Title "Role of DNA-PKcs in secondary brain injury following intracerebral hemorrhage" is not optimal because it does not reflect effects in detail.
2. the indicators do not contain detailed mechanisms because the inhibitor of pathway is not in use.
3. Administration of NU7441, a selective inhibitor of DNA-PKcs, alleviated neurological impairment, histological damage, and ipsilateral brain edema in vivo. Mechanistically, NU7441 attenuated neuronal apoptosis both in vivo and in vitro, alleviated oxidative stress by decreasing ROS levels, and suppressed ferroptosis by enhancing GPX4 activity. These contents only indicate that it is detrimental in acute brain injury.
Thus, the study is devoid of enough depth.

Validity of the findings

1. Title "Role of DNA-PKcs in secondary brain injury following intracerebral hemorrhage" is not optimal because it does not reflect effects in detail.
2. the indicators do not contain detailed mechanisms because the inhibitor of pathway is not in use.
3. Administration of NU7441, a selective inhibitor of DNA-PKcs, alleviated neurological impairment, histological damage, and ipsilateral brain edema in vivo. Mechanistically, NU7441 attenuated neuronal apoptosis both in vivo and in vitro, alleviated oxidative stress by decreasing ROS levels, and suppressed ferroptosis by enhancing GPX4 activity. These contents only indicate that it is detrimental in acute brain injury.
Thus, the study is devoid of enough depth.

Additional comments

1. Title "Role of DNA-PKcs in secondary brain injury following intracerebral hemorrhage" is not optimal because it does not reflect effects in detail.
2. the indicators do not contain detailed mechanisms because the inhibitor of pathway is not in use.
3. Administration of NU7441, a selective inhibitor of DNA-PKcs, alleviated neurological impairment, histological damage, and ipsilateral brain edema in vivo. Mechanistically, NU7441 attenuated neuronal apoptosis both in vivo and in vitro, alleviated oxidative stress by decreasing ROS levels, and suppressed ferroptosis by enhancing GPX4 activity. These contents only indicate that it is detrimental in acute brain injury.
Thus, the study is devoid of enough depth.

---

## Round 0.2 · accepted · Accept

Upon evaluation of your comprehensive reply, the reviewers have deemed the changes to your manuscript adequate, and feel it is ready for publication.

·

Basic reporting

No comment.

Experimental design

No comment.

Validity of the findings

No comment.

Additional comments

I thank the authors for addressing the issues in my initial review. I am satisfied with the additional sentences added to the manuscript and have no additional suggestions to make